# Change in population structure, policy adjustment, and China's public pension sustainability

Weiqiang Fan[1], Jia Meng[2]*, Hualei Yang[3]

1 School of Economics and Management, East China Jiaotong University, Nanchang, China, 2 School of Economics and Management, Wuhan Sports University, Wuhan, China, 3 School of Public Administration, Zhongnan University of Economics and Law, Wuhan, China

* mengjia@stu.zuel.edu.cn

## Abstract

This study assesses the impact of population structure changes and policy adjustment on public pension sustainability. The analysis is based on actuarial models for pension income, expenditure, and accumulated balance, assessed under varying scenarios. Based on the results, China's pension financial situation will be in deficit by around 2028, with accumulated deficit potentially as high as RMB 147,411.037 billion in 2050 without population policy adjustment. Second, matching the fertility level with replacement level will only slightly ease the pension financial situation after 2041 and cannot change the deficit trend or the time of first appearance of deficit. The short-term situation may worsen slightly. Third, delaying the legal retirement age to 65 years significantly improves the pension financial situation and ensures no accumulated pension deficit before 2050. Lastly, although increasing economic growth and reducing pension growth can significantly improve the pension financial situation, if the accumulated pension deficit does not appear before 2050, the pension growth rate should be controlled below 0 and economic growth must double in future. In conclusion, to improve the financial status of pensions and ensure elderly welfare, we should focus on reducing pension growth, increasing economic growth, and postponing the statutory retirement age.

## 1. Introduction

Most developed countries entered the aging society in the middle of the 20th century; the adverse effects of population aging on the economy and social security system attracted the attention of several scholars [1–3]. Previous studies [4]pointed out that the aging population in Germany will cause the pension insurance system to bear huge repayment pressure and even generate funding gaps. Some researchers [5] predicted that the ratio of US pension insurance expenditure to GDP will rise from 8% in 1999 to 21% in 2075 with aging. It has been argued that population ageing under

**Data availability statement:** All relevant data are within the paper and its Supporting information files.

**Funding:** 21FRKB003/The Post-funded Projects of the National Social Science Fund.

pay-as-you-go will result in an unsustainable pension system for most OECD countries [6]. Some researchers [7–12] use other countries as samples and corroborate the inference that "population aging will reduce the sustainability of basic pension funds." Scholars have also actively explored the impact of China's aging population on pension funds. It has been [13] pointed out that given the context of population aging and the pay-as-you-go fund arrangement, if no institutional reform measures are undertaken, the Chinese public pension fund will have a current deficit by 2025. Some researchers [14,15] believes that a quarter of the world's elderly will live in China in 2030 and that the population aged over 65 years will account for 22% of China's total population. If the pension system is not adjusted accordingly, China will use the equivalent of 10% of the same period's GDP to subsidize the pension gap. Some researchers [16] used the actuarial model to find that China's basic pension insurance fund will have a current gap in 2030 and will expand annually thereafter. If a pension gap is achieved through the contribution rate, the latter will reach 37%. As China's population ages, Some researchers [17] predicted that the size of China's pension fund gap will reach 2.023 trillion yuan in 2020 and 731.319 billion yuan by 2050. Some researchers [18] further corroborates the above points.

In order to mitigate the adverse effects of population aging on pension funds, scholars have actively investigated potential policy responses. Among these, the effects of fertility policy adjustments have generated considerable debate. At the international level, it has been [19] advocated restoring fertility rates to the replacement level to maintain a balanced population structure, while some researchers [20], using a demographic simulation model, demonstrated that a two-child policy would be more effective than a one-child policy in reducing pension deficits. Similarly, previous studies [21], based on an overlapping generations (OLG) model, found that increasing fertility could alleviate the financial pressure on public pension systems, though only when fertility reached a certain threshold. Studies by Chinese scholars reveal that the effects of fertility policies are characterized by significant time lags and regional heterogeneity. Some researchers [22] simulated the impact of the selective two-child policy and found that it could postpone the onset of pension deficits but could not reverse the long-term trend of deficit expansion. Some researchers [23] argued that although the universal three-child policy might increase the future contribution base, in the short run it could reduce current contribution capacity due to a decline in female labor force participation.

Research on strategies to enhance pension sustainability has primarily focused on two major reform directions: delayed retirement and the optimization of contribution parameters. First, regarding delayed retirement, extending the retirement age could improve pension fund balances by prolonging contribution periods and shortening benefit durations [24]. However, its effectiveness could be constrained by the benefit replacement effect and potential crowding-out effects in the labor market [25]. Chinese empirical studies show that delayed retirement significantly improves pension sustainability during the first 15 years of implementation, but the positive effects gradually diminish over a 30-year horizon, exhibiting an initial increase followed by a subsequent decline [26,27]. Second, regarding optimization of contribution parameters,

transitioning from a pay-as-you-go (PAYG) system to a multi-pillar pension system has become an international consensus [28,29]. Some researchers [30] suggested lowering the basic pension replacement rate to address demographic changes, whereas another researchers [31] criticized the current benefit adjustment mechanism for excessive administrative intervention, which could exacerbate long-term payment risks.

Although these studies have made significant achievements, there are at least four aspects to expand on. First, considering that urban employee pension insurance is the most important component of China's social pension insurance system, based on the Chinese sample, there have been scarce studies that investigate the impact of changes in population structure on the sustainable development of urban employee pension insurance funds in China. Second, when scholars study the impact of fertility on pension funds, they usually neglect the impact of fertility on actuarial parameters including payment base and contribution base, thereby ignoring the short-term effects of the fertility policy. Third, there is little international work to measure the impact of China's delayed retirement on pension funds. Fourth, there is a relative lack of attention to economic growth and the effect of adjusting the treatment mechanism in public policies to deal with the impact of population structural changes on pension funds.

Based on the above shortcomings, this study establishes an actuarial model to simulate the impact of changes in population structure on urban employee pension insurance funds during the critical period of realizing two hundred-year goals in China. Using this model, we answer three streams of questions: First, we address scientific questions such as whether there will be a current deficit and accumulated deficit in China's pension fund before 2050, and if so, when; second, whether the financial situation of pension funds can be improved if the fertility level reaches the replacement level in the future or the plan of delaying retirement is adopted; finally, should the pension growth rate be controlled and to what extent or by how much will economic growth minimally increase if there is no accumulated deficit in future pension funds.

## 2. Model building and parameter setting

The Decision of the State Council on Establishing a Unified Basic Pension Insurance System for Corporate Employees (State Council Document No. 26, 1997), promulgated on July 16, 1997, marked the establishment of China's basic pension insurance system for urban employees. Subsequently, the Decision of the State Council on Improving the Basic Pension Insurance System for Corporate Employees (State Council Document No. 38, 2005), issued on December 3, 2005, partially amended the provisions of the 1997 decision. Notably, the two decisions adopt different classification criteria for participants in the basic pension insurance system, resulting in inconsistencies across policy stages. For analytical clarity and to facilitate comparative research, this study reclassifies participants into four distinct groups based on their policy exposure and benefit calculation rules. Pre-1997 retirees (Pre-1997 retirces refer to those retired before 1997), post-1997 retirees(Post-1997 retirees refer to those retired between 1998 and 2005), post-2005 retirees(Post-2005 retirees refer to those retired after 2006), and post-1997 employees(Post-1997 employees refer to those slart working from 1998). China's Urban Employee Basic Pension (UEBP) system adopts a partially funded model that combines social pooling with individual accounts. The social pooling component operates on a pay-as-you-go (PAYG) basis, while the individual accounts are fully funded, functioning essentially as personal savings and therefore not directly relevant to sustainability concerns. Accordingly, in this paper, the term "pension fund" refers exclusively to the social pooling component of the public pension system. The actuarial analysis of the UEBP in this study covers the period from 2020 to 2050, representing a medium- to short-term simulation of pension fund sustainability.

### 2.1 Pension income model

The social pension fund income in year $t$ equals the number of current employees who pay the social pension insurance in that year multiplied by the corresponding insurance contribution base, and by the insurance contribution ratio in year $t$. This can be expressed as follows:

$$AI_t = [\sum_{i=1}^{4} \sum_{x=a_t}^{b_t-1} N_{t,x}^i] \times \overline{w}_t \times R_t = [\sum_{i=1}^{4} \sum_{x=a_t}^{b_t-1} N_{t,x}^i] \times \overline{w}_{t_0-1} \times \prod_{s=t_0}^{t} (1+k_s) \times R_t \tag{1}$$

where, $AI_t$ represents the pension income; $i$ is the individual identity variable such that $i=1$ for pre-1997 retirees, $i=2$ for post-1997 retirees, $i=3$ for post-2005 retirees, and $i=4$ for post-1997 employees; $a_t$ and $b_t$ respectively denote the age of participation in social pension insurance and the retirement age in year $t$; $\sum_{i=1}^{4} N_{t,x}^i$ denotes the total number of current employees participating in social pension insurance in year $t$; $\overline{w}_t$ denotes the contribution base in year $t$; $t_0$ is the start time of actuarial analysis (i.e., 2020); $k_t$ denotes growth rate of this contribution base in year $t$; and $R_t$ denotes the contribution ratio in year $t$.

## 2.2 Pension expenditure model

$AC_t$ denotes the social pension fund expenditure in year $t$, which includes basic pension fund expenditure, $AC_{t,b}$, and transitional pension expenditure, $AC_{t,g}$, in year $t$.

The basic pension fund expenditure in year $t$ equals the total number of retired employees multiplied by the per capita basic pension in year $t$. In turn, the per capita basic pension in year $t$ equals the basic pension calculation base multiplied by the calculation ratio and the growth rate. This can be expressed as follows:

$$AC_{t,b} = \sum_{i=1}^{4} \sum_{x=b_t}^{c_t} [N_{t,x}^i \times \overline{B}_{t,x}^i \times s_{t,x}^i \times \prod_{s=t-x+b_t}^{t} (1+g_s)] \tag{2}$$

where $c_t$ is the maximum survival age of $i$-class workers in year $t$, $\overline{B}_{t,x}^i$ and $s_{t,x}^i$ respectively denote the pension calculation base and calculation ratio for $i$-class employees in year $t$, and $g_t$ and $g_t + 1$ respectively denote the basic pension growth rate and growth coefficient.

Transitional pension expenditure in year $t$ equals the number of post-1997 and post-2005 retirees in year $t$ multiplied by the per capita transitional pension in that year. In turn, the latter is given by the transitional pension calculation base multiplied by the deemed payment period, transitional pension calculation ratio, and growth rate. It can be expressed as follows:

$$AC_{t,g} = \sum_{i=2}^{3} \sum_{x=b_t}^{c_t} \{N_{t,x}^i \times \overline{G}_{t,x}^i \times [1998 - (t-x+a_t)] \times v_{t,x}^i \times \prod_{s=t-x+b_t}^{t} (1+g_s)\} \tag{3}$$

where $\overline{G}_{t,x}^i$ is the transitional pension calculation base of $i$-class workers in year $t$; $[1998-(t-x+a_t)]$ ($\geq 0$)denotes the deemed payment period of $i$-class workers in year $t$; $v_{t,x}^i$ denotes the transitional pension calculation ratio of $i$-class workers in year $t$; and the growth rate of the transitional pension in year $t$ equals the growth rate of the base pension in year $t$.

## 2.3 Pension accumulated balance model

The accumulated balance of the pension in year $t$ equals the accumulated balance of the pension in year $t$-1 plus the current balance of the pension fund in year $t$. In turn, the latter equals the income minus the expenditure of the pension insurance in year $t$. This can be expressed as follows:

$$F_t = F_{t-1}(1+r) + (AI_t - AC_t) \tag{4}$$

The r represents the one-year fixed deposit interest rate offered by banks.

## 2.4 Projection of urban employee pension contributors

The age-specific population distribution in urban and rural areas from 2020 to 2050 is projected using the cohort-component method. Specifically: The population by age and by urban/rural residence in year t equals the corresponding population in year t–1 multiplied by the age-specific survival probability. The number of newborns in year t equals the sum of the product of the female population aged 15–49 years and the age-specific fertility rates in that year. By incorporating the urbanization rate, the age-specific urban and rural population distribution can be derived.

Next, the future number of urban employees enrolled in the basic pension system is projected: Baseline year assumption (2019): It is assumed that the age distribution of insured urban employees mirrors that of the total urban employed population, and the age distribution of urban retirees enrolled in the pension system corresponds to that of the total urban retired population. For example, in 2019, urban employees aged 20 accounted for 2.119% of the total urban employed population. Given that the total number of insured urban employees was 262.192 million, the number of insured 20-year-old employees was 5.556 million (262.192 million × 2.119%). Repeating this calculation yields the number of insured employees aged 20–54 and the number of retired employees aged 55–100 in 2019.

Dynamic projection (2020–2050): Using the cohort-component method, the number of urban employees contributing to the pension system and those receiving pension benefits is estimated annually by age group for 2020–2050. Each year, employees aged 100 years are removed from the system, while new employees aged 20 years are added, ensuring the dynamic turnover of the insured population.

## 2.5 Parameter setting

In terms of labor and retirement age, the Labor Law stipulates that the minimum age for working is not less than 16 years. However, considering the extension of education, it is assumed that the age at first employment of urban employees is 20 years. At present, the retirement age is 60 years for male workers, 50 years for female workers, and 55 years for female cadres. It is assumed that the average retirement age of urban employees is 55 years. Based on the sixth census, the relaxation of the fertility policy in China, and World Bank estimates, China's current fertility level is set at about 1.6.

Regarding the contribution rate of social pension insurance, enterprises pay 20% and individuals pay 8%. The contribution base of social pension insurance is the average wage of the employees in the previous year. In terms of contribution base and pension growth rate, the former is mainly affected by wages, and wages are determined by the market. Therefore, the growth rate of the pension contribution base is directly linked with the economic growth rate. Due to the different economic growth rates in the two fertility scenarios, the growth rate of the contribution base also differs. This economic growth rate in the two fertility scenarios (presented in Table 1) mainly refers to the previous papers [32]. While the pension growth rate is controlled by the government, the latter's pension regulation is mostly based on the economic growth rate. If the pension growth rate is not adjusted in the high-fertility situation, it remains set on the basis of the economic growth rate in the benchmark fertility situation; otherwise, it is set based on the economic growth rate under the high-fertility scenario.

According to State Council Document No. 26 of 1997 and No. 38 of 2005, the basic pension of pre-1997 and post-1997 retirees accounts for 70% and 20% of the contribution base, respectively. Using the calculation ratio of the basic pension of post-2005 and post-1997 retirees, and each additional year in pension contributions, the basic pension and transitional pension calculation ratios increased by 1% and 1.2%, respectively.

**Table 1. GDP growth rate at different fertility rates in 2020–2050.**

| GDP growth rate | 2020 | 2021–2025 | 2026–2030 | 2031–2035 | 2036–2040 | 2041–2045 | 2046–2050 |
|---|---|---|---|---|---|---|---|
| Benchmark (TFR = 1.6) | 6.6 | 5.633 | 4.983 | 4.54 | 3.935 | 3.151 | 2.474 |
| High fertility (TFR = 1.94) | 6.493 | 5.296 | 4.513 | 4.344 | 4.132 | 3.519 | 2.832 |

## 3. Results: Change in population structure and pension sustainability

### 3.1 Changes in the population structure of urban employees in China under population aging

Accurately predicting future trends in the urban employee population is fundamental for estimating the future pension revenues and expenditures. Based on the aforementioned population projection method and calibrated with relevant real-world parameters, this study first simulates the trends in the total urban employee population, the number of pension contributors, the number of pension beneficiaries, and the system dependency ratio up to 2050. The results are presented in Table 2.

According to Table 2, several key trends can be observed: First, regarding the total urban employee population, the overall trend shows a gradual increase throughout the projection period, albeit with a slowing growth rate. Under the current fertility level, the total urban employee population is expected to peak at 449.4653 million in 2045, after which it will begin to decline. Second, in terms of the number of pension contributors, the current fertility level indicates a continuous downward trend, decreasing from 260.949 million in 2020 to 199.9065 million in 2050. Third, regarding the number of pension beneficiaries, the current fertility level suggests a steady increase, rising from 115.4971 million in 2020 to 245.1591 million in 2050. Fourth, as for the system dependency ratio, it exhibits an upward trend under the current fertility level, increasing from 0.44 in 2020 to 1.23 in 2050.

### 3.2 Financial performance of the pension system under changes in population structure

The impact of a change in population structure on pension sustainability is analyzed based on the above theoretical model and the results are presented in Table 3.

According to Table 3 and Fig 1, it can be observed that, first, the current pension balance shows a downward trend under a change in population structure, while the accumulated pension balance shows a slight upward trend followed by a sharp downward trend. Second, the expected time interval for the current pension balance deficit is 2022–2050 and for

Table 2. Changes in the population structure of urban employees in China.

| Year | Total Urban Employee Population | Number of Pension Contributors | Number of Pension Beneficiaries | System Dependency Ratio | Year | Total Urban Employee Population | Number of Pension Contributors | Number of Pension Beneficiaries | System Dependency Ratio |
|---|---|---|---|---|---|---|---|---|---|
| | 10,000 persons | 10,000 persons | 10,000 persons | 10,000 persons | | 10,000 persons | 10,000 persons | 10,000 persons | 10,000 persons |
| 2020 | 37644.62 | 26094.90 | 11549.71 | 0.44 | 2036 | 43475.72 | 24093.41 | 19382.31 | 0.80 |
| 2021 | 38099.83 | 25956.63 | 12143.19 | 0.47 | 2037 | 43765.49 | 23905.7 | 19859.79 | 0.83 |
| 2022 | 38517.82 | 25897.68 | 12620.14 | 0.49 | 2038 | 44020.81 | 23801.87 | 20218.94 | 0.85 |
| 2023 | 38931.84 | 25623.65 | 13308.20 | 0.52 | 2039 | 44248.65 | 23668.68 | 20579.97 | 0.87 |
| 2024 | 39371.10 | 25465.35 | 13905.75 | 0.55 | 2040 | 44443.73 | 23519.98 | 20923.75 | 0.89 |
| 2025 | 39806.50 | 25249.10 | 14557.40 | 0.58 | 2041 | 44609.81 | 23247.11 | 21362.71 | 0.92 |
| 2026 | 40226.43 | 25116.77 | 15109.66 | 0.60 | 2042 | 44750.77 | 22837.74 | 21913.03 | 0.96 |
| 2027 | 40632.00 | 24995.00 | 15637.00 | 0.63 | 2043 | 44850.62 | 22459.02 | 22391.6 | 1.00 |
| 2028 | 41016.15 | 24850.17 | 16165.98 | 0.65 | 2044 | 44915.51 | 22010.19 | 22905.32 | 1.04 |
| 2029 | 41387.69 | 24719.49 | 16668.20 | 0.67 | 2045 | 44946.53 | 21503.06 | 23443.47 | 1.09 |
| 2030 | 41682.32 | 24570.73 | 17111.59 | 0.70 | 2046 | 44924.26 | 21193.58 | 23730.68 | 1.12 |
| 2031 | 42007.36 | 24465.41 | 17541.95 | 0.72 | 2047 | 44865.86 | 20891.81 | 23974.05 | 1.15 |
| 2032 | 42329.74 | 24448.49 | 17881.25 | 0.73 | 2048 | 44776.2 | 20558.74 | 24217.46 | 1.18 |
| 2033 | 42629.08 | 24364.34 | 18264.74 | 0.75 | 2049 | 44661.28 | 20270.91 | 24390.36 | 1.20 |
| 2034 | 42922.08 | 24270.34 | 18651.74 | 0.77 | 2050 | 44506.56 | 19990.65 | 24515.91 | 1.23 |
| 2035 | 43171.77 | 24164.11 | 19007.66 | 0.79 | | | | | |

**Table 3. Financial status of pension under a change in population structure.**

| Year | Income (RMB 100 million) | Expenditure (RMB 100 million) | Current balance (RMB 100 million) | Accumulated balance (RMB 100 million) |
|------|------|------|------|------|
| 2020 | 42717.40 | 41305.53 | 1411.87 | 52866.48 |
| 2021 | 45295.45 | 45217.32 | 78.14 | 52944.62 |
| 2022 | 47738.28 | 49019.37 | −1281.09 | 51663.52 |
| 2023 | 49893.78 | 53852.87 | −3959.09 | 47704.44 |
| 2024 | 52378.71 | 58716.00 | −6337.29 | 41367.15 |
| 2025 | 54859.34 | 64153.94 | −9294.60 | 32072.55 |
| 2026 | 57645.86 | 69181.78 | −11535.92 | 20536.63 |
| 2027 | 60224.97 | 74427.12 | −14202.16 | 6334.48 |
| 2028 | 62859.62 | 80021.02 | −17161.40 | −10826.93 |
| 2029 | 65644.87 | 85852.91 | −20208.04 | −31034.97 |
| 2030 | 68501.22 | 91778.40 | −23277.18 | −54312.15 |
| 2031 | 71606.38 | 97611.99 | −26005.61 | −80317.76 |
| 2032 | 74805.55 | 103297.37 | −28491.82 | −108809.58 |
| 2033 | 77932.56 | 109544.78 | −31612.22 | −140421.80 |
| 2034 | 81156.37 | 116202.76 | −35046.40 | −175468.19 |
| 2035 | 84469.52 | 123078.14 | −38608.62 | −214076.81 |
| 2036 | 88046.08 | 129770.82 | −41724.74 | −255801.55 |
| 2037 | 90797.74 | 137521.13 | −46723.40 | −302524.95 |
| 2038 | 93960.75 | 144908.89 | −50948.13 | −353473.08 |
| 2039 | 97111.61 | 152726.34 | −55614.72 | −409087.80 |
| 2040 | 100298.87 | 160857.03 | −60558.16 | −469645.97 |
| 2041 | 103036.18 | 168927.29 | −65891.11 | −535537.08 |
| 2042 | 104411.29 | 178269.90 | −73858.61 | −609395.69 |
| 2043 | 105915.26 | 187473.27 | −81558.01 | −690953.69 |
| 2044 | 107069.32 | 197443.45 | −90374.13 | −781327.83 |
| 2045 | 107898.36 | 208095.19 | −100196.84 | −881524.66 |
| 2046 | 109696.41 | 215609.21 | −105912.80 | −987437.46 |
| 2047 | 110809.72 | 222989.26 | −112179.55 | −1099617.01 |
| 2048 | 111740.83 | 230621.44 | −118880.61 | −1218497.61 |
| 2049 | 112902.22 | 237825.21 | −124922.99 | −1343420.60 |
| 2050 | 114095.79 | 244785.56 | −130689.76 | −1474110.37 |

the accumulated pension balance deficit is 2028–2050. Third, the current pension balance is expected to decrease from RMB 141.187 billion in 2020 to RMB 13,068.976 billion in deficit in 2050. The accumulated balance of pension is expected to increase from RMB 5,286.648 billion in 2020 to RMB 5,294.462 billion in 2021, followed by an expansion in deficit to RMB 147,411.037 billion in 2050.

## 4. Results: Policy adjustment and pension sustainability

### 4.1 Financial status of pension under different fertility scenarios

In order to analyze the impact of fertility policy on the financial status of social pension, two fertility scenarios were set up. One is the benchmark fertility scenario under the current fertility level while the other is the high fertility scenario under China's universal two-child policy.

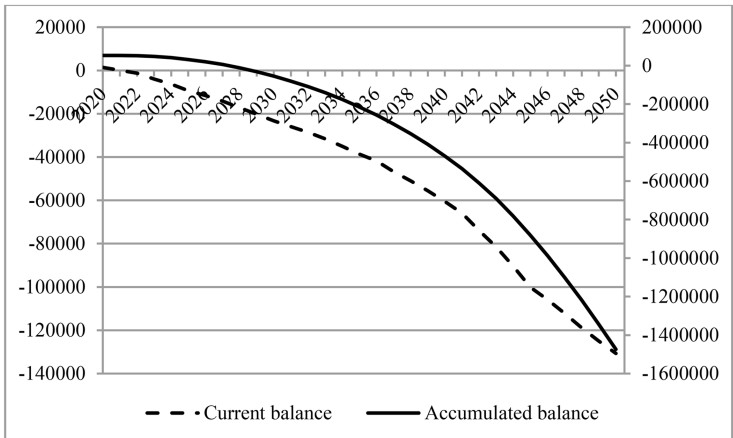

**Fig 1. Financial performance of the pension system under changes in population structure (Unit: RMB 100 million).**

According to Table 4, it can firstly be observed that the high-fertility scenario is not expected to change the trend of a slight increase followed by a rapid decline in the accumulated pension balance. Second, the high fertility scenario is not expected to change the time node of 2021 and 2028, when the current deficit and accumulated deficit first appeared in the benchmark scenario, respectively. Third, before 2041, high fertility is expected to worsen the financial status of pensions with an average annual increase in accumulated deficit of RMB 769.285 billion. After 2041, the high-fertility scenario is likely to greatly improve the pension financial status with the average annual deficit reducing by RMB 7,562.718 billion. According to Table 4, under the benchmark scenario, the accumulated pension balance will increase from RMB 5,265.106 billion in 2020 to RMB 5,294.462 billion in 2021. Thereafter, the accumulated pension deficit is expected to continue increasing. Specifically, the accumulated deficit would occur from 2028 to 2050 and would increase to RMB 147,411.037 billion in 2050. Under the high fertility scenario, the accumulated balance of pension is expected to increase from RMB 5,265.106 billion in 2020. It is expected to exhibit a downward trend from 2021 onwards until there would be an accumulated deficit for the first time in 2028, with the accumulated pension deficit expanding to RMB 129,721.132 billion in 2050.

### 4.2 Financial Status of Pensions under Different Delayed Retirement Scenarios

Referring to the provisions of the Labor Law of the People's Republic of China and some researchers [33] setting of retirement scenarios, we analyze the impact of delayed retirement on the financial status of pensions by setting up three retirement scenarios. First, there is no delay in retirement under the benchmark scenario with the retirement age set at 55 years. In the second scenario, there is an immediate postponement in retirement age such that the individuals who have not yet retired would only retire at the age of 65. The third scenario involves a gradual postponement in retirement age by six months each year to eventually reach the 65 years mark.

According to Table 5, first, by and large, delayed retirement is expected to change the declining trend of the accumulated pension balance under the benchmark scenario, while the corresponding graph under the delayed retirement scenario has an inverted U-shape that rises first and then decreases slightly. Second, delayed retirement delays the time at which accumulated pension deficit first appears under the benchmark scenario. Third, delayed retirement is expected to improve the financial situation of pensions, and the improvement is greater under the immediately delayed retirement scenario than under the gradually delayed retirement scenario. Before 2050, the accumulated pension deficit was expected to be reduced by an average of RMB 80,435.282 billion per year under the immediately delayed retirement scenario and by an average of RMB 63,698.256 billion per year under the gradually delayed retirement scenario.

**Table 4. Financial operation of pension under different fertility situations.**

| Year | Current balance(RMB 100 million) | | Accumulated Balance (RMB 100 million) | |
|---|---|---|---|---|
| | Benchmark scenario | High fertility scenario | Benchmark scenario | High fertility scenario |
| 2020 | 1411.87 | 1302.26 | 52866.48 | 52651.06 |
| 2021 | 78.14 | −70.98 | 52944.62 | 52580.08 |
| 2022 | −1281.09 | −1570.17 | 51663.52 | 51009.91 |
| 2023 | −3959.09 | −4381.20 | 47704.44 | 46628.71 |
| 2024 | −6337.29 | −6898.39 | 41367.15 | 39730.32 |
| 2025 | −9294.60 | −9989.03 | 32072.55 | 29741.30 |
| 2026 | −11535.92 | −12373.45 | 20536.63 | 17367.85 |
| 2027 | −14202.16 | −15248.73 | 6334.48 | 2119.12 |
| 2028 | −17161.40 | −18412.12 | −10826.93 | −16293.00 |
| 2029 | −20208.04 | −21662.49 | −31034.97 | −37955.49 |
| 2030 | −23277.18 | −24937.58 | −54312.15 | −62893.06 |
| 2031 | −26005.61 | −27875.64 | −80317.76 | −90768.71 |
| 2032 | −28491.82 | −30408.83 | −108809.58 | −121177.54 |
| 2033 | −31612.22 | −33545.73 | −140421.80 | −154723.27 |
| 2034 | −35046.40 | −36978.65 | −175468.19 | −191701.91 |
| 2035 | −38608.62 | −40529.10 | −214076.81 | −232231.01 |
| 2036 | −41724.74 | −42559.89 | −255801.55 | −274790.91 |
| 2037 | −46723.40 | −45980.87 | −302524.95 | −320771.78 |
| 2038 | −50948.13 | −48573.45 | −353473.08 | −369345.23 |
| 2039 | −55614.72 | −51495.87 | −409087.80 | −420841.10 |
| 2040 | −60558.16 | −54601.42 | −469645.97 | −475442.51 |
| 2041 | −65891.11 | −57977.90 | −535537.08 | −533420.41 |
| 2042 | −73858.61 | −63707.17 | −609395.69 | −597127.58 |
| 2043 | −81558.01 | −69136.35 | −690953.69 | −666263.93 |
| 2044 | −90374.13 | −75610.20 | −781327.83 | −741874.13 |
| 2045 | −100196.84 | −83027.89 | −881524.66 | −824902.02 |
| 2046 | −105912.80 | −86412.94 | −987437.46 | −911314.96 |
| 2047 | −112179.55 | −90416.24 | −1099617.01 | −1001731.20 |
| 2048 | −118880.61 | −94824.18 | −1218497.61 | −1096555.38 |
| 2049 | −124922.99 | −98593.89 | −1343420.60 | −1195149.26 |
| 2050 | −130689.76 | −102062.06 | −1474110.37 | −1297211.32 |

According to Table 5, the accumulated pension balance is likely to increase from RMB 5,286.648 billion in 2020 to RMB 5,294.462 billion in 2021 under the benchmark scenario. Thereafter, the accumulated balance is expected to exhibit a downward trend with an accumulated deficit occurring from 2028 to 2050. Under the immediately delayed retirement scenario, the accumulated pension balance is expected to increase from RMB 8,694.240 billion in 2020 to RMB 70,853.115 billion in 2045, subsequently, showing a downward trend, although the accumulated deficit would likely not appear before 2050. Under gradually delayed retirement, the accumulated pension balance is expected to increase from RMB 5,286.648 billion in 2020 to RMB 52,132.641 billion in 2047 before showing a downward trend subsequently, although no accumulated deficit is expected before 2050.

### 4.3 Financial Status of Pension under Different Pension Growth Rates

In order to analyze the impact of different pension growth rates on the financial status of pension, three pension scenarios are set up. First, under the benchmark scenario, pension growth rate is set to be equal to the corresponding economic

**Table 5. Financial Operation of Pension under Different Delayed Retirement Scenarios.**

| Year | Current balance(RMB 100 million) | | | Accumulated Balance (RMB 100 million) | | |
|------|-----------------------|------------------------------|---------------------------|-----------------------|------------------------------|---------------------------|
| | Benchmark scenario | Immediately Delayed retirement | Gradual Delayed retirement | Benchmark scenario | Immediately Delayed retirement | Gradually Delayed retirement |
| 2020 | 1411.87 | 16268.42 | 8951.86 | 52866.48 | 86942.40 | 67551.31 |
| 2021 | 78.14 | 19850.53 | 7993.61 | 52944.62 | 106792.93 | 75544.91 |
| 2022 | −1281.09 | 23288.69 | 10795.55 | 51663.52 | 130081.62 | 86340.47 |
| 2023 | −3959.09 | 27100.30 | 9120.71 | 47704.44 | 157181.91 | 95461.17 |
| 2024 | −6337.29 | 31372.35 | 12452.75 | 41367.15 | 188554.27 | 107913.92 |
| 2025 | −9294.60 | 36087.61 | 10833.83 | 32072.55 | 224641.88 | 118747.75 |
| 2026 | −11535.92 | 41463.79 | 15033.15 | 20536.63 | 266105.67 | 133780.90 |
| 2027 | −14202.16 | 41739.29 | 13992.35 | 6334.48 | 307844.96 | 147773.25 |
| 2028 | −17161.40 | 40010.89 | 18361.43 | −10826.93 | 347855.85 | 166134.68 |
| 2029 | −20208.04 | 38918.72 | 15516.32 | −31034.97 | 386774.57 | 181650.99 |
| 2030 | −23277.18 | 37260.96 | 20266.46 | −54312.15 | 424035.54 | 201917.45 |
| 2031 | −26005.61 | 35930.26 | 18093.66 | −80317.76 | 459965.80 | 220011.11 |
| 2032 | −28491.82 | 35117.36 | 23561.03 | −108809.58 | 495083.16 | 243572.14 |
| 2033 | −31612.22 | 31995.35 | 20246.31 | −140421.80 | 527078.51 | 263818.45 |
| 2034 | −35046.40 | 29491.44 | 26390.89 | −175468.19 | 556569.95 | 290209.35 |
| 2035 | −38608.62 | 25995.42 | 23738.96 | −214076.81 | 582565.37 | 313948.31 |
| 2036 | −41724.74 | 24011.54 | 31330.33 | −255801.55 | 606576.91 | 345278.64 |
| 2037 | −46723.40 | 21415.88 | 28864.13 | −302524.95 | 627992.79 | 374142.77 |
| 2038 | −50948.13 | 18468.42 | 26028.83 | −353473.08 | 646461.21 | 400171.60 |
| 2039 | −55614.72 | 15531.76 | 23186.40 | −409087.80 | 661992.97 | 423358.00 |
| 2040 | −60558.16 | 13008.69 | 20734.75 | −469645.97 | 675001.65 | 444092.75 |
| 2041 | −65891.11 | 11318.53 | 19029.46 | −535537.08 | 686320.18 | 463122.21 |
| 2042 | −73858.61 | 9647.53 | 17314.04 | −609395.69 | 695967.71 | 480436.25 |
| 2043 | −81558.01 | 7086.21 | 14672.29 | −690953.69 | 703053.92 | 495108.54 |
| 2044 | −90374.13 | 4137.27 | 11604.99 | −781327.83 | 707191.19 | 506713.53 |
| 2045 | −100196.84 | 1339.96 | 8652.79 | −881524.66 | 708531.15 | 515366.32 |
| 2046 | −105912.80 | −1177.79 | 5894.09 | −987437.46 | 707353.36 | 521260.41 |
| 2047 | −112179.55 | −6727.95 | 65.99 | −1099617.01 | 700625.41 | 521326.41 |
| 2048 | −118880.61 | −10722.62 | −4240.74 | −1218497.61 | 689902.80 | 517085.67 |
| 2049 | −124922.99 | −15067.72 | −8929.02 | −1343420.60 | 674835.07 | 508156.65 |
| 2050 | −130689.76 | −19494.52 | −13726.31 | −1474110.37 | 655340.55 | 494430.34 |

growth rate. Second, the annual pension growth rate is decreased by 50% compared to the benchmark scenario. Third, the annual pension growth rate equals 0, which guarantees no accumulated deficit before 2050.

According to Table 6, first, reducing pension growth rate delays the time when the accumulated deficit first appears compared to the benchmark scenario. Second, reducing pension growth rate improves the financial status of the pension; the lower the pension growth rate, the better is the pension financial status. If the pension growth rate was reduced by 50%, the accumulated deficit of pension is expected to reduce by RMB 28,175.049 billion annually on average before 2050, and if the pension growth rate is 0, there is expected to be no accumulated deficit until 2050.

According to Table 6, when the pension growth rate is decreased by 50%, the accumulated pension balance is expected to increase from RMB 5,286.648 billion in 2020 to RMB 7,382.357 billion in 2025, and subsequently, show a

**Table 6. Financial status of pension under different pension growth rates.**

| Year | Current balance(RMB 100 million) | | | Accumulated Balance (RMB 100 million) | | |
|------|----------------------|------------------------------------|--------------------------|----------------------|------------------------------------|--------------------------|
| | Benchmark scenario | Pension growth rate slowed by 50% | Pension growth rate is 0 | Benchmark scenario | Pension growth rate slowed by 50% | Pension growth rate is 0 |
| 2020 | 1411.87 | 4678.43 | 7752.61 | 52866.48 | 59152.78 | 65184.61 |
| 2021 | 78.14 | 4479.92 | 8515.88 | 52944.62 | 63632.71 | 73700.49 |
| 2022 | −1281.09 | 4358.41 | 9402.17 | 51663.52 | 67991.11 | 83102.66 |
| 2023 | −3959.09 | 3015.65 | 9104.95 | 47704.44 | 71006.77 | 92207.61 |
| 2024 | −6337.29 | 2096.67 | 9292.95 | 41367.15 | 73103.43 | 101500.56 |
| 2025 | −9294.60 | 720.14 | 9078.76 | 32072.55 | 73823.57 | 110579.32 |
| 2026 | −11535.92 | −39.48 | 9375.65 | 20536.63 | 73784.10 | 119954.97 |
| 2027 | −14202.16 | −1123.95 | 9394.44 | 6334.48 | 72660.15 | 129349.41 |
| 2028 | −17161.40 | −2398.80 | 9270.38 | −10826.93 | 70261.35 | 138619.79 |
| 2029 | −20208.04 | −3652.04 | 9219.99 | −31034.97 | 66609.31 | 147839.77 |
| 2030 | −23277.18 | −4813.41 | 9316.45 | −54312.15 | 61795.90 | 157156.22 |
| 2031 | −26005.61 | −5763.38 | 9514.10 | −80317.76 | 56032.52 | 166670.32 |
| 2032 | −28491.82 | −6381.00 | 10088.51 | −108809.58 | 49651.52 | 176758.83 |
| 2033 | −31612.22 | −7554.76 | 10140.20 | −140421.80 | 42096.76 | 186899.03 |
| 2034 | −35046.40 | −8952.23 | 10012.79 | −175468.19 | 33144.53 | 196911.82 |
| 2035 | −38608.62 | −10385.50 | 9897.34 | −214076.81 | 22759.03 | 206809.16 |
| 2036 | −41724.74 | −11724.29 | 9644.22 | −255801.55 | 11034.74 | 216453.38 |
| 2037 | −46723.40 | −14882.44 | 7613.59 | −302524.95 | −3847.70 | 224066.97 |
| 2038 | −50948.13 | −17186.20 | 6496.91 | −353473.08 | −21033.90 | 230563.88 |
| 2039 | −55614.72 | −19861.14 | 5058.25 | −409087.80 | −40895.04 | 235622.13 |
| 2040 | −60558.16 | −22733.88 | 3478.20 | −469645.97 | −63628.92 | 239100.33 |
| 2041 | −65891.11 | −26691.17 | 388.34 | −535537.08 | −90320.08 | 239488.67 |
| 2042 | −73858.61 | −33246.16 | −5252.81 | −609395.69 | −123566.25 | 234235.86 |
| 2043 | −81558.01 | −39482.95 | −10514.08 | −690953.69 | −163049.19 | 223721.78 |
| 2044 | −90374.13 | −46778.09 | −16770.17 | −781327.83 | −209827.28 | 206951.61 |
| 2045 | −100196.84 | −55027.29 | −23916.13 | −881524.66 | −264854.57 | 183035.48 |
| 2046 | −105912.80 | −59937.72 | −28190.32 | −987437.46 | −324792.29 | 154845.16 |
| 2047 | −112179.55 | −65400.66 | −32994.12 | −1099617.01 | −390192.95 | 121851.04 |
| 2048 | −118880.61 | −71305.34 | −38221.51 | −1218497.61 | −461498.29 | 83629.54 |
| 2049 | −124922.99 | −76551.78 | −42768.32 | −1343420.60 | −538050.07 | 40861.21 |
| 2050 | −130689.76 | −81535.83 | −47041.04 | −1474110.37 | −619585.90 | −6179.82 |

declining trend. While this would not result in a deficit until 2037, by 2050, the accumulated deficit will likely be as high as RMB 61,958.59 billion. Next, when the pension growth rate is zero, the accumulated pension balance is expected to increase from RMB 5,286.648 billion in 2020 to RMB 23,948.867 billion in 2041; subsequently, the expected accumulated pension balance shows a declining trend, but there is no accumulated deficit until 2050.

## 4.4 Financial status of pension under different economic growth

In order to analyze the impact of economic growth rates on pension financial status, three scenarios of economic growth are set up here. First, under the benchmark scenario, the economic growth rate remains unchanged. Second, compared with the benchmark scenario, the current economic growth rate is increased by 50.3%. Third, compared with the

benchmark scenario, the economic growth rate is increased by 100.6%, which guarantees that there would be no accumulated deficit in pensions before 2050.

According to Table 7, first, higher economic growth delays the time at which the accumulated pension deficit first appears. Second, increasing economic growth would improve the financial situation of pensions; the higher the economic growth rate, the better would be the financial situation. Third, compared with the benchmark scenario, under the condition of 50.3% increase in economic growth, the total pension deficit is likely to be reduced by RMB 29,657. 449 billion annually, on average, before 2050, while under the condition of 100.6% increase in economic growth, the total pension deficit is expected to be reduced by RMB 71,746.055 billion annually, on average, and, of course, there would be no accumulated deficit before 2050.

**Table 7. Financial operation of pension under different economic growth rates.**

| Year | Current balance (RMB 100 million) | | | Accumulated Balance (RMB 100 million) | | |
|---|---|---|---|---|---|---|
| | Benchmark scenario | Economic growth increased by 50.3% | Economic growth increased by 100.6% | Benchmark scenario | Economic growth increased by 50.3% | Economic growth increased by 100.6% |
| 2020 | 1411.87 | 9690.33 | 15693.11 | 52866.48 | 76218.08 | 90120.30 |
| 2021 | 78.14 | 9469.79 | 17215.20 | 52944.62 | 85687.87 | 107335.49 |
| 2022 | −1281.09 | 9796.92 | 19580.61 | 51663.52 | 95484.79 | 126916.11 |
| 2023 | −3959.09 | 8648.24 | 20428.91 | 47704.44 | 104133.04 | 147345.01 |
| 2024 | −6337.29 | 7981.17 | 22084.39 | 41367.15 | 112114.20 | 169429.41 |
| 2025 | −9294.60 | 6694.11 | 23221.71 | 32072.55 | 118808.31 | 192651.11 |
| 2026 | −11535.92 | 5642.61 | 24531.52 | 20536.63 | 124450.92 | 217182.63 |
| 2027 | −14202.16 | 4567.54 | 26038.28 | 6334.48 | 129018.46 | 243220.91 |
| 2028 | −17161.40 | 3183.72 | 27378.74 | −10826.93 | 132202.18 | 270599.65 |
| 2029 | −20208.04 | 1774.32 | 28950.74 | −31034.97 | 133976.50 | 299550.39 |
| 2030 | −23277.18 | 431.11 | 30910.85 | −54312.15 | 134407.61 | 330461.24 |
| 2031 | −26005.61 | −1118.37 | 32438.31 | −80317.76 | 133289.25 | 362899.55 |
| 2032 | −28491.82 | −1775.22 | 35564.44 | −108809.58 | 131514.03 | 398464.00 |
| 2033 | −31612.22 | −3313.07 | 37693.65 | −140421.80 | 128200.96 | 436157.65 |
| 2034 | −35046.40 | −5256.05 | 39530.47 | −175468.19 | 122944.91 | 475688.12 |
| 2035 | −38608.62 | −7331.95 | 41498.37 | −214076.81 | 115612.95 | 517186.48 |
| 2036 | −41724.74 | −10177.78 | 41606.40 | −255801.55 | 105435.17 | 558792.88 |
| 2037 | −46723.40 | −15352.90 | 38133.14 | −302524.95 | 90082.28 | 596926.02 |
| 2038 | −50948.13 | −19248.00 | 36961.39 | −353473.08 | 70834.28 | 633887.41 |
| 2039 | −55614.72 | −23925.15 | 34701.88 | −409087.80 | 46909.13 | 668589.28 |
| 2040 | −60558.16 | −29117.52 | 31790.20 | −469645.97 | 17791.61 | 700379.49 |
| 2041 | −65891.11 | −37830.80 | 20983.42 | −535537.08 | −20039.19 | 721362.91 |
| 2042 | −73858.61 | −50172.05 | 3602.54 | −609395.69 | −70211.24 | 724965.45 |
| 2043 | −81558.01 | −62292.49 | −13594.95 | −690953.69 | −132503.73 | 711370.50 |
| 2044 | −90374.13 | −76777.02 | −35337.44 | −781327.83 | −209280.76 | 676033.06 |
| 2045 | −100196.84 | −93527.21 | −61621.15 | −881524.66 | −302807.97 | 614411.91 |
| 2046 | −105912.80 | −105427.71 | −81176.69 | −987437.46 | −408235.68 | 533235.22 |
| 2047 | −112179.55 | −117293.17 | −100969.27 | −1099617.01 | −525528.85 | 432265.95 |
| 2048 | −118880.61 | −130316.13 | −123330.87 | −1218497.61 | −655844.98 | 308935.08 |
| 2049 | −124922.99 | −142314.76 | −144032.24 | −1343420.60 | −798159.74 | 164902.84 |
| 2050 | −130689.76 | −154058.14 | −164575.01 | −1474110.37 | −952217.89 | 327.84 |

According to Table 7, when the economic growth rate is increased by 50.3%, the accumulated deficit pension balance increases from RMB 7,621.808 million in 2020 to RMB 13,440.761 billion in 2030; subsequently, it shows a declining trend such that there is no deficit until 2041 but, by 2050, the accumulated deficit would be as high as RMB 95,221.789 billion. When the economic growth rate is increased by 100.6%, the cumulative pension balance is expected to increase from RMB 9,012.030 billion in 2020 to RMB 72,496.545 billion in 2042; subsequently, the accumulated pension balance shows a declining trend, but there is no accumulated deficit before 2050.

## 5. Conclusions

Based on the actuarial model and taking the basic pension insurance of urban employees in China as an example, this study focuses on the impact of changes in population structure and population policy adjustments on pension sustainability in the critical period of realizing China's "two hundred-year goals."

The conclusions are as follows: (1) With a change in population structure, China's pension financial situation is likely to show a deteriorating trend such that a pension accumulated deficit would first appear in 2028 and would be as high as RMB 147,411.037 billion in 2050. (2) Compared with the benchmark scenario, the high fertility scenario would improve the financial situation of pensions after 2041 while slightly worsening the situation before 2041. However, even if the fertility level matched the replacement level, it would not change the trend of pension accumulated deficit after 2028. (3) Delayed retirement significantly improves the financial situation of pensions; the greater is the intensity of delayed retirement, the better is its effect. Generally speaking, it results in an increasing trend in the accumulated balance of pensions with no deficit before 2050. (4) Reducing the pension growth rate would help to improve the financial situation of pensions; the lower is the pension growth rate, the better is the pension financial situation. If the pension growth rate is equal to or below 0, there would be no accumulated deficit until 2050. (5) The higher the economic growth rate compared to the benchmark scenario, the better is the pension financial situation. If the future economic growth is more than double that of the benchmark scenario, there would be no deficit in pensions before 2050.

In summary, given China's trend of rapid aging in the future, in order to improve the financial status of pensions and protect the welfare of the elderly, we should focus on reducing pension growth and increasing economic growth, and especially postponing the statutory retirement age. Nonetheless, this study has some limitations. For example, it does not consider the status of pension funds in other fertility scenarios, the impact of retirement policies on actuarial parameters, and the effect of gender differences on pension funds. These areas could form topics for future research.

## Supporting information

**S1 Data. Support data.**
(XLSX)

## Author contributions

**Conceptualization:** Weiqiang Fan, Hualei Yang.

**Data curation:** Weiqiang Fan, Jia Meng.

**Formal analysis:** Weiqiang Fan.

**Funding acquisition:** Hualei Yang.

**Software:** Jia Meng.

**Writing – original draft:** Jia Meng.

**Writing – review & editing:** Jia Meng.

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
