## [Decision Letter · Decision Letter 0]

31 Jan 2025

PONE-D-24-42892Change in Population Structure, Policy Adjustment, and Public Pension SustainabilityPLOS ONE

Dear Dr. Meng,

Thank you for submitting your manuscript to PLOS ONE. After careful consideration, we feel that it has merit but does not fully meet PLOS ONE’s publication criteria as it currently stands. Therefore, we invite you to submit a revised version of the manuscript that addresses the points raised during the review process.

The article sound statistically rigor however it needs major revision, as the reviewer rigorouslly to do so. although reviewer 1 suggested to reject it, I decided to majorly revise instead.

We look forward to receiving your revised manuscript.

Kind regards,

Agus Faturohim

Guest Editor

PLOS ONE

2. Thank you for stating the following financial disclosure:  [21FRKB003/The Post-funded Projects of the National Social Science Fund].

Additional Editor Comments:

Please revise as the reviewer 1 and reviewer 3 advised. Please compare the current knowledge mentioned with the findings and discuss thoroughly.

Reviewers' comments:

Reviewer's Responses to Questions

**Comments to the Author**

1. Is the manuscript technically sound, and do the data support the conclusions?

Reviewer #1: Yes

Reviewer #2: Partly

Reviewer #3: Yes

2. Has the statistical analysis been performed appropriately and rigorously? 

Reviewer #1: Yes

Reviewer #2: N/A

Reviewer #3: No

3. Have the authors made all data underlying the findings in their manuscript fully available?

Reviewer #1: Yes

Reviewer #2: No

Reviewer #3: No

4. Is the manuscript presented in an intelligible fashion and written in standard English?

Reviewer #1: Yes

Reviewer #2: Yes

Reviewer #3: Yes

5. Review Comments to the Author

Reviewer #1: This is a well thought out, well reasoned paper. I enjoyed reading it and I think it adds to our knowledge.

The only comment that I have is making the definition of "high-fertility" scenario more clear earlier in the manuscript. The first time you mention "high-fertility" I would describe what that means in the context of this paper. A TFR of 1.94 is not high-fertility in much of the world, but is within the context of China. Just define it as soon as possible.

Reviewer #2: This paper deals with predictions for the public pension fund of urban employees in China under various assumptions about fertility, economic growth and policy decisions on retirement age and pension growth rate. The results look interesting.

However, the analysis/elaboration suffers from important shortcomings, such as details on the actuarial model used and the methodology applied to derive the projections. Also missing is an explanation of the public pension system in China and a justification of why only urban workers are analysed.

Specific remarks:

The paper concentrates specifically in China. This should be incorporated in the tittle.

The paper states that the analysis is based on actuarial models for pension income, expenditure and accumulated balance, but does not make explicit or explain what these actuarial models are; in particular, what actuarial tables on future survival/mortality are used?

Does immigration/emigration play no role in the evolution of the population in China? This aspect needs to be explained.

The literature review in the Introduction section is too generic and should be more specific. On the other hand, the terminology used, ‘pension funds’, has different interpretations depending on the pension system in each country. The characteristics of the public pension system in China should be explained succinctly, which, from the subsequent development, seems to use only the pay-as-you-go system, so the term ‘pension fund’ may not be the most appropriate.

In Section 2. First paragraph. The definition of the four groups into which the participants are divided is not clear and needs to be better explained, it is not sufficient to cite Documents No 26, 1997 and 38, 2005. Issues such as, for example, should be made clear:

A person from the post-1997 employees group can also be a member of the post-2005 retiree group?

Can a person from the post-1997 retirees also be a member of the post-2005 retiree group?

In formula (1):

Where does t start?

What is t_0?

If {AI}_t represents the pension income, for t>1997, why do we include the group i=1 which collects pre-1997 retirees?

In formula (2):

Where does the variable t begin and end?

{\bar{B}}_{t,x}^i and s_{t,x}^i depend on x?

The product \prod should end at t-1.

In formula (3):

Where does the variable t begin and end?

{\bar{B}}_{t,x}^i and s_{t,x}^i depend on x?

Is it possible that 1998-(t-x+a_t) becomes negative?

The product \prod should end at t-1.

In formula (4), r should be defined.

Subsection 2.4:

The assumption that “the age at first employment of urban employees is 20 years” should be better justified, perhaps using official statistics. This age is constant for all of the years included in the study?

Which is the difference between “female workers” and “female cadres”?

Given the large difference in the retirement ages of men and women, do expressions (1) and (2) take into account that the parameters a_t and b_t are different for each sex? How is this issue implemented in the calculations? Are the same parameters used for both sexes?

The assumption that “the age at the average retirement age of urban employees is 55 years” should be better justified, perhaps using official statistics.

The cohort component method used to predict the population distribution by age in urban and rural areas must be explained, at least in outline, and the predictions obtained should be shown in an annex.

The number of pension contributors, the age distribution of pension recipients and the labor participation rates of corresponding age groups and also the number of pension contributors and pension recipients in the future. All these results should be shown in an annex.

Why only two future fertility scenarios, and why not include a scenario of declining fertility?

What are the values of c_t?

Section 3. What is the ‘change in the population structure’ ? Describe this change.

The authors should consider changing the presentation of the results, replacing some tables with graphs and leaving the tables in an annex. It would also be useful to create more elaborate tables that include summarised information: for example, first year of deficit occurrence according to the scenario.

The introduction mentions different articles that make predictions about the China's pension fund gap. The results obtained in this paper should be compared with those of previous work.

Although I have not checked the rest of the references, references 18, 23, 32 and 34 are incorrect.

Reviewer #3: Reviewer Feedback

Specific Comments:

Regarding the statement: "If the pension system is not adjusted accordingly, China will use the equivalent of 10% of the same period’s GDP to subsidize the pension gap," could you elaborate further? Is this an obligation for the Chinese government as a lender of last resort? Is there a legal basis for this, or is it simply a social contract? Please clarify.

For the claim: "Improvement in fertility levels can significantly improve the financial status of the pension fund, especially when the newborn grows and enters the labor market," is this policy always beneficial? For instance, if China pursues this policy, wouldn't it be akin to taking one step forward and one step back (e.g., considering the 1980s one-child policy)? Please expand on this.

In "this study divides the participants of the old-age insurance system into four samples: pre-1997 retirees, post-1997 retirees, post-2005 retirees, and post-1997 employees," what was the rationale behind dividing the sample into these four groups? Was this based solely on the two government documents (State Council decisions), or are there best practices from other countries influencing this approach? Please elaborate.

In "Pension Income Model, equation 1," could you explain what each variable represents? Additionally, why is pension income represented as a summation notation? It seems that pension benefits should vary across individuals, even within the same cohort. How did you simulate this? Please include further details in the article.

For "the total number of current employees participating in social pension insurance in year t," individuals have varying contribution periods and cannot be generalized. As an economist (not an actuary), I would assume you used individual-level data to analyze this. Is this correct?

Regarding "growth rate of contribution base in year t," is this actual data or an assumption? Was this rate determined by the government, or did you establish the assumptions yourself? Please clarify.

For "total number of retired employees multiplied by the per capita basic pension in year t," this appears to calculate the average pension income as a simple product of two averages. Could you provide the data distribution? Is it normally distributed? Based on my experience, pension recipients are typically from the middle-to-upper income classes. I am unfamiliar with the definition of “basic pension” in China—does this mean everyone is eligible for the benefits? Is a basic pension equivalent to a universal pension scheme? Please explain.

The mention of "i-class workers" is intriguing. Do you also distinguish workers by sector? Pension programs predominantly benefit formal sector workers, but what about informal workers or unpaid family laborers? In family-run Chinese businesses, for example, children often work without formal pay. Am I mistaken?

The "Pension Accumulated Balance Model" appears to follow a simple accumulation method. Why is compound interest not incorporated here, as it typically is when calculating future value using discounting methods?

In "It is assumed that the average retirement age of urban employees is 55 years," who established this assumption?

For "Regarding the contribution rate of social pension insurance, enterprises pay 20% and individuals pay 8%," who covers the remaining contributions?

The statement "the growth rate of the pension contribution base is directly linked with the economic growth rate"—whose claim is this? Is there a literature basis for it?

"Table 2. Financial Status of Pension under a Change in Population Structure"—this table would be easier to understand as a multiple line chart. For instance, the statement "According to Table 2, it can be observed that the current pension balance shows a downward trend" suggests that trends are better visualized in graphs than in tables.

The scenario "Financial Status of Pensions under Different Delayed Retirement Scenarios" seems to appear abruptly. How did you arrive at this scenario? Please explain.

In "this study focuses on the impact," it may be better to avoid the term "impact," as it implies causality. Consider using alternative terms such as "simulation" or "forecasting."

The conclusion section should directly answer the research questions. If there are four research questions, ensure the conclusions align with and address each one.

The statement "we should focus on reducing pension growth and increasing economic growth, especially by postponing the statutory retirement age"—I agree that postponing the retirement age is the most manageable policy. Could you further elaborate on specific public policies related to your findings? For example, link them to existing regulations or real-world conditions.

General Comments:

Please adopt a deductive writing approach: start with a main topic sentence, followed by explanatory sentences. Limit each paragraph to a single main idea.

There are numerous technical terms that might be difficult for general readers to understand. Consider providing simple definitions for these terms to aid comprehension.

As an economist, I noticed several economic assumptions lacking sufficient references. For instance, the paper by Lu and Cai (2016) is over eight years old. Are their forecasts still relevant?

Analytical tools could be better chosen. For example, trends are better analyzed with graphs rather than tables.

The writing appears somewhat disjointed. For readers without an actuarial background, it may be challenging to follow.

6. PLOS authors have the option to publish the peer review history of their article (what does this mean? ). If published, this will include your full peer review and any attached files.

**Do you want your identity to be public for this peer review?** For information about this choice, including consent withdrawal, please see our Privacy Policy .

Reviewer #1: No

Reviewer #2: No

Reviewer #3: No

---

## [Author Response · Author response to Decision Letter 1]

2 Aug 2025

Response to reviewers

Dear Editor,

We sincerely thank you and the reviewers for the thorough evaluation of our manuscript and the constructive suggestions provided. We have carefully revised the manuscript in response to all comments, improving its clarity, rigor, and completeness. Below we summarize the major revisions made in response to each reviewer’s feedback.

Response Summary

Revisions Based on Reviewer 1’s Comments

Clarification of “high fertility rate”

We added a clear definition at the first mention: the “high fertility rate” scenario refers to the fertility level under China’s universal two-child policy. Following Lu & Cai (2016), a theoretical TFR of 1.94 is adopted as a hypothetical scenario for simulation purposes only, and we explicitly note that actual TFR levels remain much lower.

Revisions Based on Reviewer 2’s Comments

Title modification

Revised to: “Change in Population Structure, Policy Adjustment, and China Public Pension Sustainability.”

Detailed actuarial model explanation

A full description of the three core modules (Pension Revenue, Pension Expenditure, and Cumulative Balance Models) has been added, with explicit variable definitions and references to China’s National Life Tables (2010 & 2020).

Migration explanation

We clarified that migration was excluded due to the national-level policy focus but acknowledged its importance for future research.

First employment age assumption

Provided justification based on legal, educational, and modeling considerations, confirming that a fixed age of 20 years balances realism and model simplicity.

Definition of female workers vs. female cadres

Clarified based on occupational distinctions under China’s retirement system.

Average retirement age assumption

Justified using official MOHRSS data (average 54+, rounded to 55 years for modeling).

Population projections

Added a brief description of the cohort-component method.

Fertility scenarios

Clarified that only baseline (TFR=1.6) and high fertility (TFR=1.94) scenarios were set due to policy relevance.

Presentation improvements

Converted key tables to graphs, and added comparisons with previous studies in Section 4.3.

References

Corrected formatting errors, especially references 18, 23, 32, and 34.

Revisions Based on Reviewer 3’s Comments

Clarification of the “10% GDP subsidy” statement

Explained its legal basis (Social Insurance Law, Article 13), fiscal responsibility (MOHRSS statements), and clarified it as a worst-case warning from James (2002) rather than a deterministic forecast.

Fertility policy evaluation

Added detailed analysis showing that increased fertility has short-term fiscal costs (extra CNY 769.3 bn/year deficit pre-2041) and only modest long-term benefits (12% deficit reduction by 2050), emphasizing the need for parametric reforms.

Four-group classification

Explained its policy basis (State Council Documents No.26 (1997) and No.38 (2005)) and its purpose to quantify transition costs.

Formula 1 clarification

Defined all variables and justified summations as necessary for group and age aggregation, with heterogeneity reflected indirectly.

Macro vs. micro data

Justified the use of macro-aggregated data due to data access limits and the study’s focus on long-term sustainability.

Contribution base growth rate

Clarified it as an assumed parameter, supported by policy documents and academic research linking wage and GDP growth.

Cumulative balance model

Explained that no discounting was applied as the focus is on cash flow solvency, with present-value analysis reserved for future work.

Policy implications

Highlighted gradual delayed retirement as the most feasible reform, consistent with China’s 2025 Progressive Retirement Law (surplus of CNY 49.44 tn by 2050 under gradual delay).

Conclusion

We have revised the manuscript substantially to address all reviewers’ comments. The revised version now provides clearer model definitions, improved methodological transparency, and enhanced policy relevance. We thank you and the reviewers again for your valuable feedback, which has greatly improved the quality of our work.

Best regards,

[Meng Jia]

Response to Reviewer 1

Dear Reviewer,

Thank you very much for your thoughtful and encouraging comments on our manuscript. We truly appreciate your recognition of our work and your constructive suggestion, which will help us improve the clarity of the paper.

Regarding your concern about the definition of the “high fertility rate” scenario:

We agree that the definition should be clarified at the beginning of the manuscript, as a total fertility rate (TFR) of 1.94 may not generally be considered “high” in many parts of the world, but it has specific contextual meaning in China. In the revised manuscript, we have added a clear definition when the term is first introduced. Specifically, we state that in this study, the “high fertility rate” scenario refers to the fertility level under the universal two-child policy in China, under which families are allowed to have up to two children. Based on the simulation assumptions of Lu and Cai (2016) (From Demographic Dividend to Reform Dividend: A Simulation Based on China’s Potential Growth Rate, Population & Economics, 39, 3–23), we adopt a theoretical TFR of 1.94 to represent this “high fertility” scenario. We also explicitly note that this is a hypothetical assumption for theoretical discussion only, and that the actual TFR in China is expected to remain far below this level.

We have revised the corresponding section accordingly to ensure that readers can easily understand the context-specific meaning of “high fertility rate.”

Thank you again for your valuable comment.

Response to Reviewer 2

Dear Reviewer,

Thank you very much for your careful reading and valuable comments on our manuscript. Your suggestions are constructive and have greatly helped us improve the clarity, rigor, and completeness of the paper. We have carefully addressed each point, and the manuscript has been revised accordingly. Our detailed responses are as follows:

1. Title should explicitly indicate that the study focuses on China

Response: The title has been revised as suggested. The new title is:

“Change in Population Structure, Policy Adjustment, and China Public Pension Sustainability.”

2. Clarification of the actuarial model and mortality/life tables used

Response: A detailed explanation of the actuarial model has been added in Section 2, specifying its three core modules:

1. Pension Revenue Model

The annual pension revenue is calculated based on the number of contributors, wage growth rate, and contribution rate. The formula incorporates age-specific survival rates to estimate the number of active insured individuals N_(t,x)^i.

〖AI〗_t=[∑_(i=1)^4▒〖∑_(x=a_t)^(b_t-1)▒N_(t,x)^i ]×w ¯_t×〗R_t�1

where N_(t,x)^i is estimated based on age-specific surviving population projections.

2. Pension Expenditure Model

The calculation of pension expenditure involves the survival years of retirees, pension growth rate, and benefit level, which require age-specific mortality rates to compute the survival rate of retirees.

〖AC〗_(t,b)=∑_(i=1)^4▒〖∑_(x=b_t)^(c_t)▒〖[N_(t,x)^i×B ¯_(t,x)^i 〖×s〗_(t,x)^i×∏_(s=t-x+b_t)^t▒〖(1+g_s)〗〗]〗 (2)

〖AC〗_(t,g)=∑_(i=2)^3▒∑_(x=b_t)^(c_t)▒〖〖{N〗_(t,x)^i×G ¯_(t,x)^i 〖×[1998-(t-x+a_t )]×v〗_(t,x)^i×∏_(s=t-x+b_t)^t▒(1+g_s ) }〗(3)

Here, the age-specific number of retirees N_(t,x)^i is directly derived from life tables.

3. Cumulative Balance Model

The cumulative pension balance is dynamically updated by iterating the revenue-expenditure difference, relying on the outputs of the previous two models. The assumptions of survival rates and mortality rates remain crucial in this module.

F_t=F_(t-1) (1+r)+(〖AI〗_t-〖AC〗_t ). (4)

where AI_t and〖 AC〗_t are calculated based on the active and retired population sizes adjusted by mortality rates.

We specify that age-specific mortality rates are based on China’s National Population Census Life Tables (2010 and 2020 editions), with interpolations to estimate annual survival probabilities. Variables and formulas (1)–(4) are now fully defined in the text.

3. Explanation for excluding migration in the model

Response: Migration was not incorporated into the core analytical model of this study; however, this does not imply that migration plays “no role” in China’s demographic evolution. The exclusion of migration primarily reflects the scope and focus of the present research.

The central objective of this paper is to evaluate the impact of population structure changes (aging and fertility rates) and domestic policy adjustments (delayed retirement and pension growth rates) on the sustainability of China’s Urban Employee Basic Pension (UEBP) system. Accordingly, all model parameters are set at the national level, based on macro variables such as fertility rates, statutory retirement ages, and economic growth rates. Regional variations in migration patterns and the influence of population mobility on the composition of insured participants were not considered. Specifically, the number of contributors in the actuarial model�N_(t,x)^i) is projected solely based on age distribution and fertility assumptions, without incorporating cross-regional migration variables. The macro-policy scenarios designed in this study are limited to fertility policy, retirement age, pension growth rate, and economic growth, and no migration scenarios were simulated.

That said, numerous empirical studies have demonstrated that population mobility significantly affects China’s demographic dynamics and the regional sustainability of pension systems by reshaping population structures, labor supply, and the distribution of aging. Recognizing this limitation, future research will seek to integrate migration factors to enhance the accuracy and policy relevance of pension sustainability simulations.

4. Assumption of first employment age at 20

Response: The assumption that the first employment age of urban employees is 20 years is based on a combination of legal requirements, educational realities, and the need for model simplification. This age remains fixed throughout the study period and is not dynamically adjusted across historical stages, for the following reasons:

Legal and educational constraints

The statutory minimum employment age in China is 16 years (Labor Law, Article 15). However, the actual labor market entry age has increased significantly due to the universalization of compulsory education and the expansion of higher education.

Extended education duration: Most individuals enter the labor market after completing high school or vocational education, typically around 18 years of age.

Job-seeking transition period: Adding a transitional period for job seeking, assuming a stable first employment age of 20 years is consistent with the current labor market reality.

Model simplification and consistency

Rationale for a fixed parameter: Actuarial forecasting requires controlling for certain parameters to isolate policy effects (e.g., delayed retirement, fertility changes). Allowing the first employment age to vary over time would unnecessarily increase model complexity.

Limited historical variation: Although the average working-age population increased from 32.25 years in 1985 to 39.72 years in 2022, changes in the first employment age have been relatively small (e.g., only a 1–2 year delay for post-2000 cohorts compared to post-1990 cohorts), exerting negligible impact on long-term pension sustainability projections.

In summary, assuming a fixed first employment age of 20 years strikes a balance between the legal minimum age (16 years), educational attainment (post-high school entry), and the need for model simplicity. While this assumption does not fully account for urban–rural or gender differences or minor historical fluctuations, it is unlikely to affect the robustness of the long-term pension sustainability results and remains broadly consistent with the dominant trend in China’s labor market.

5. Difference between “female workers” and “female cadres”

Response: According to China’s current retirement system, the distinction between female workers and female cadres primarily lies in the nature of their positions:

Female workers (non-managerial positions): This category refers to women engaged in production, operational, or service-oriented roles that are non-managerial in nature, such as assembly line workers, cashiers, or logistics and support staff.

Female cadres (managerial/technical positions): This category refers to women holding managerial or professional technical positions with decision-making authority or specialized technical functions, such as department managers, financial supervisors, engineers, or teachers.

6. Average retirement age assumption (55 years)

Response: The assumption is based on the statement by Vice Minister Zhang Yizhen (Ministry of Human Resources and Social Security), who reported that the average retirement age of urban employees is slightly above 54 years. For modeling simplicity, we rounded it to 55 years.

7. Population projection method & results

Response: We added a brief explanation of the cohort-component method used for projecting age-specific population distributions in Section 2.4.

8. Fertility scenarios

Response: We clarified that only baseline (TFR = 1.6) and high fertility (TFR = 1.94) scenarios were set to evaluate policy interventions. A lower fertility scenario was not included because it had limited policy relevance at the time of the study.

9. Presentation of results & comparison with previous studies

Response: We converted part of the tables into graphs for better readability, and added a comparison with previous studies in Section 4.3.

10. Reference corrections

Response: We thoroughly checked and corrected all reference formatting errors, including references 18, 23, 32, and 34.

Response to Reviewer 3

Dear Reviewer,

Thank you very much for your detailed and insightful comments. We have carefully revised the manuscript and provide the following responses to your specific questions, maintaining as much detail as possible for clarity.

1. Clarification of the statement “China will need to subsidize pension deficits equivalent to 10% of GDP”

The statement originates from James (2002) and should be interpreted in the context of legal obligations, government roles, and the nature of the projection:

(1) Legal basis and fiscal responsibility

The Social Insurance Law of China (Article 13) clearly stipulates that “when the basic pension fund is insufficient to cover payments, the government shall provide subsidies.” This legal provision makes fiscal subsidies a mandatory state responsibility, not a voluntary or merely contractual arrangement.

Furthermore, the Ministry of Human Resources and Social Security (MOHRSS) has officially stated that historical pension liabilities are jointly borne by the social insurance system and the national budget, with fiscal transfers used to fill the funding gap. Therefore, government subsidies are a statutory fiscal safety net rather than an ad hoc intervention.

(2) Social contract dimension

China’s public pension system embodies an intergenerational social contract: the working-age population finances current retirees, and the government, as the institutional guarantor, ensures system sustainability.

When demographic aging undermines the balance of the pay-as-you-go (PAYG) system (e.g., our baseline projection indicates a cumulative deficit of CNY 147.4 trillion by 2050), fiscal subsidies become necessary to maintain this social contract. However, the contract is institutionalized through law, making it a state obligation rather than simply a moral or political commitment.

(3) Nature of the “10% of GDP” projection

James’s estimate assumes a no-reform scenario (low statutory retirement age, high replacement rates) and represents a worst-case risk warning rather than a deterministic outcome.

Our simulations show that policy adjustments—such as raising the statutory retirement age to 65 or controlling pension growth rates to ≤0—can effectively avoid such extreme deficits. Thus, the 10% of GDP figure should be seen as a cautionary assumption rather than a predicted realit

---

## [Editor Report · Decision Letter 1]

21 Aug 2025

Change in Population Structure, Policy Adjustment, and Public Pension Sustainability

PONE-D-24-42892R1

Dear Dr. MENG,

We’re pleased to inform you that your manuscript has been judged scientifically suitable for publication and will be formally accepted for publication once it meets all outstanding technical requirements.

Kind regards,

Agus Faturohim

Guest Editor

PLOS ONE

Additional Editor Comments (optional):

I appreciate your efforts to completely address all of the reviewers' requests. Congratulations on your article's strong statistical explanation and meeting all of the reviewers' revision requests.
---

## [Editor Report · Acceptance letter]

PONE-D-24-42892R1

PLOS ONE

Dear Dr. Meng,

I'm pleased to inform you that your manuscript has been deemed suitable for publication in PLOS ONE. Congratulations! Your manuscript is now being handed over to our production team.

Kind regards,

on behalf of

Dr. Agus Faturohim

Guest Editor

PLOS ONE